# Coarse Technogenic Material in Urban Surface Deposited Sediments (USDS)

**Andrian Seleznev** [1,2,*], **Ekaterina Ilgasheva** [1], **Ilia Yarmoshenko** [1] **and Georgy Malinovsky** [1]

1 Institute of Industrial Ecology, Ural Branch of the Russian Academy of Sciences, 20, S. Kovalevskoy Str., 620219 Ekaterinburg, Russia; boomo4ka@mail.ru (E.I.); ivy@ecko.uran.ru (I.Y.); georgy@ecko.uran.ru (G.M.)
2 Ural Federal University Named after the First President of Russia B.N. Yeltsin, 19, Mira Str., 620002 Ekaterinburg, Russia
* Correspondence: sandrian@rambler.ru

**Abstract:** In the current paper, the analysis of heavy mineral concentrate (Schlich analysis) was used to study the particles of technogenic origin in the samples of urban surface-deposited sediments (USDS). The USDS samples were collected in the residential areas of 10 Russian cities located in different economic, climatic, and geological zones: Ufa, Perm, Tyumen, Chelyabinsk, Nizhny Tagil, Magnitogorsk, Nizhny Novgorod, Rostov-on-Don, Murmansk, and Ekaterinburg. The number of technogenic particles was determined in the coarse particle size fractions of 0.1–0.25 and 0.25–1 mm. The types of technogenic particle were studied by scanning electron microscopy (SEM) analysis. The amount of technogenic material differed from city to city; the fraction of technogenic particles in the samples varied in the range from 0.01 to 0.43 with an average value of 0.18. The technogenic particles in USDS samples were represented by lithoid and granulated slag, iron and silicate microspheres, fragments of brick, paint, glass, plaster, and other household waste. Various types of technogenic particle differed in morphological characteristics as well as in chemical composition. The novelty and significance of the study comprises the following: it has been shown that technogenic particles are contained in a significant part of the USDS; the quantitative indicators of the accumulation of technogenic particles in the urban landscape have been determined; the contributions of various types of particles to the total amount of technogenic material were estimated for the urban landscape; the trends in the transformation of typomorphic elemental associations in the urban sediments associated with the material of technogenic origin were demonstrated; and the alteration trends in the USDS microelemental content were revealed, taking into account the impurities in the composition of technogenic particles.

**Keywords:** urban environment; residential area; urban surface deposited sediments; road dust; technogenic particles; slag; spherules; microplastic; plaster





## 1. Introduction

Sediment deposition in the urban area reduces the environmental quality, and affects health, aesthetics, economics, and other aspects of city life [1]. The constant sediment supply increases the costs of municipal services and cleaning the territories, as well as deteriorating urban infrastructure facilities [2–6]. The deposited loose sedimentary materials silt stormwater systems, compact urban soils, decrease the fertility of the topsoil, etc. [7–10]. The deposited solid matter on streets and sidewalks increases the wear and tear of vehicles [7–13]. Dust deposition in electrical equipment may cause outages on electricity lines [14].

Coarse sand material of road-deposited sediments is about 50% of road-deposited sediments mass [15]. The coarse particles of anthropogenic origin may contain toxic heavy metals [16–20]. The large size fraction material of road-deposited sediments (>100 μm) contains the mass of heavy metals within particulate matter similar to the fine fractions [21]. The coarse particles are involved in the transport of heavy metal pollution from roads

to stormwater drains, and they absorb pollutants and may release them during rainy periods [6,15].

The local surface geochemical traps in an urban environment representing sediment of the depressed areas of microrelief (in other words, surface dirt sediment) were chosen as the main object of the study. This environmental component is deposited on various surfaces forming the upper part of the cultural layer on the territory of the city. The sediments participate in the processes of migration and accumulation of pollutants and particulate matter.

Sediments are formed as a result of the natural processes of the weathering of the material of building constructions, pavements, and roads under freezing and thawing in the presence of moisture, soil, and ground erosion under the influence of surface stormwater runoff, and atmospheric dust deposition. The material of the excavated ground, the products of road surface abrasion by passing parking cars, and household waste also contribute to the formation of the particulate materials of the urban sediment. The accumulation of sediments significantly increases under bad cleaning conditions and poor urban management and landscaping [1,16].

USDS (urban surface-deposited sediments) is a common term characterizing the various types of loose sediment formed as a result of weathering, erosion, and destruction of soils, pavements, and construction in the urban environment, which is deposited in depressed areas as a result of surface runoff of relief [22–24]. The solid material of the USDS is composed of the particles of soil, sand, peat, dust, and small debris [25]. The formation of sediments occurs within the urban area where the various surfaces and buildings are constructed in different years and decades [18,19]. The thickness of the sediments varies within the area of the quarter and landscape functional zones and is on average 5 cm. The content of pollutants in the sediments characterizes the pollution of the area from which the sediment was accumulated [25].

The sediment includes the particles of natural and anthropogenic origin. In the urban environment, about 60% of the sediment is represented by the material of bedrock, as well as organic material [26]. Many authors have shown that ash, slag, and metal particles of various shapes and composition, metal, wear products of vehicles and other mechanisms, small household waste, as well as microplastics can be found in the composition of various types of surface sediments in an urban environment [7,11,15–18,26–28]. Organic objects in the urban surface sediment may include bacteria and viruses [29]. The technogenic material produced by the road traffic and found along the roads mainly consists of magnetic particles, which can be the products of motor vehicles: angular and spherical iron-oxides, tungsten-rich particles, and sodium chloride, with a size of about 100 μm [30].

The studies of the technogenic phase in USDS and dust in the urban environment are mainly focused on the effect of traffic on the content of particles <100 microns in size [31], in particular smaller PM2–PM10 particles due to their greatest environmental hazard and the largest accession for wind transfer and inhalation by humans [21,32,33]. Larger particles are less studied, however, and they can also hold fine dust particles on their surfaces due to electrostatic charge. Solid material from non-exhaust emissions as well as coarse material from roadway destruction, pavement abrasion, and vehicle parts are less studied [31,34]. Such loose material may be as well transferred by the wind several tens of meters away from the roadway.

Particles are redistributed between the various landscape zones by stormwater runoff, may participate in the urban sedimentary cascade entering the water bodies, and form material of bottom sediments of lakes, rivers, and estuaries [35–39]. The particles may adsorb pollutants, bacteria, and viruses. Contemporary USDS in the city is a good collector of pollutants and material of different origins, including non-point sources of pollution.

Road traffic is one of the main sources of technogenic material [30,40,41] such as the particles of wear of tires, brake pads, and road abrasion products. Tire wear products contribute the most part of anthropogenic material in road dust, galley sediments, pavement dust, car park dust, and roadside soils and snow. Anthropogenic material from vehicles is

represented by magnetic particles including spherules and slag, comprising the particles of about 100 μm size [30,32,42]. Smelters and coal-fired power plants also represent significant sources of anthropogenic solid material in cities, forming non-point sources of pollution, such as fly ash [17,43–45].

Thus, the identification of sources of anthropogenic material, the content of technogenic materials, and the assessment of the amount and types of anthropogenic particles in different parts of the landscape are among the significant environmental issues in an urban environment.

While the environmental role of the USDS in modern cities had been demonstrated in the previous studies involving such characteristics as pollution with the heavy metals [22,24,25,46] and the contribution of the dust fraction [23], this study has been focused on the technogenic particles in the urban environment. The objectives of the study were: (1) the identification of particles of the anthropogenic origin found in the urban environment compartments; (2) the classification and characterization of the morphological features of technogenic particles; (3) the assessment of the amount of technogenic material in urban surface deposited sediments; and (4) in an urban environment; and (5) the characterization of cities according to the amount of technogenic material in the contemporary urban surface sediments.

## 2. Materials and Methods

### 2.1. The Description of the Studied Cities

The USDS sample collection program was performed in 10 Russian cities located in different climatic and industrial zones, in the territories with different geological structure (Figure 1) [47]: Ufa, Perm, Tyumen, Chelyabinsk, Nizhny Tagil, Magnitogorsk, Nizhny Novgorod, Rostov-on-Don, Murmansk, and Ekaterinburg. The chosen cities have a high automobile traffic load, >250 cars per 1000 people, and high density of population.

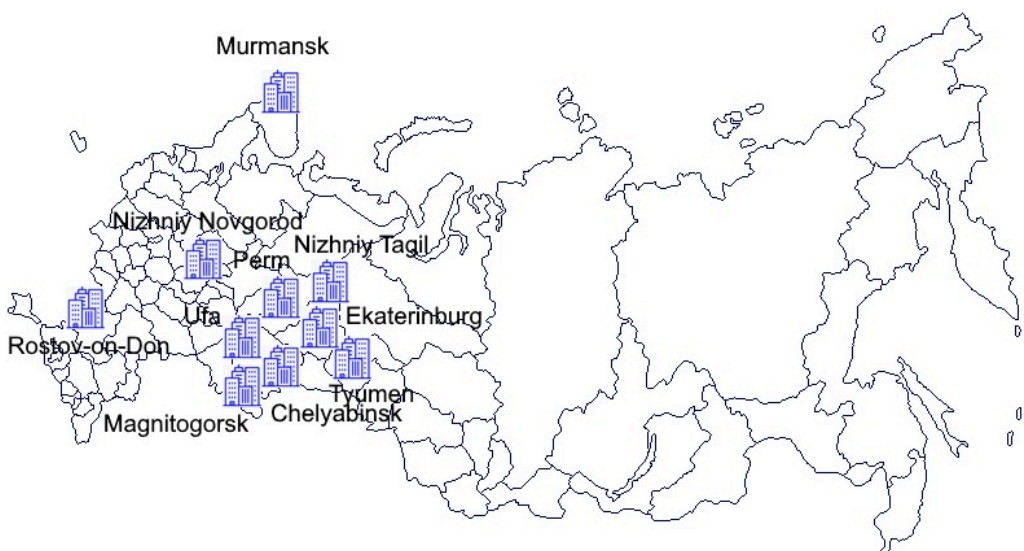

**Figure 1.** The location of the cities on the map of Russia where the collection of samples of urban surface deposited sediments was performed.

The significant development of urbanization in the cities occurred in the second half of the 20th century. The descriptions of the surveyed cities are represented in Table 1.

**Table 1.** The description of the surveyed cities.

| City, Population, Million People/Cars per 1000 People/City Area (km²) | Geographic and Climatic Zone, Average Temperature, °C, Jan/Jul | Geological Features [47] | Main Industries |
|---|---|---|---|
| Ufa, 1.1/278/707.9 | Forest-steppe zone, temperate continental climate, −12.4/19.7 | Volga-Ural Anteclise, Verkhnekamsk basin; gypsum, anhydrite, sandstone, marl, siltstone, dolomite, Pre-Jurassic limestones; alluvium, colluvium, diluvium, sandstones, sandy loams, loams, Upper Pliocene and Holocene clays. | Oil processing, oil chemical industry, machinery |
| Perm, 1.007/237/803 | Forest zone, temperate continental climate, −12.8/18.6 | East of the European part of Russia, banks of the Kama River. Pre-Ural geomorphological zone, Kungurian stage, Irvinskaya and Fillipovskaya formations of the Pre-Jurassic: gypsum, sandstone, limestones, dolomites, marls; clays, loam and sandy loam of Holocene alluvial, eluvial and diluvial sediments. | Electric power industry, oil and gas processing, machinery, chemistry and petrochemistry, woodworking. |
| Tyumen, 0.77/363/698.5 | Western Siberia, forest taiga zone with waterlogged areas, temperate continental climate, −15/18.8 | West Siberian plain, Tyumen downwarp; diorites and gabbros of the Pre-Jurassicformations; loams, clays, silts and lake-alluvium of the Upper Pliocene and Holocene. | Oil processing, gas-fired power plants |
| Chelyabinsk, 1.2/269/530 | The South Urals, forest-steppe zone, temperate climate, −14.1/19.3 | East Urals uplift and West Side of West Siberian plate; granites, diorites, coals, limestones, sandstones, dolomitic limestones of the Pre-Jurassic formations; sands, siltstones, loams, alluvial sediments of floodplain terraces, pebbles, gravels, and eluvial-diluvial sediments of the Upper Pliocene and Holocene. | Ferrous and non-ferrous metallurgy, chemical industry, machinery, coal-fired power plants |
| Nizhniy Tagil, 0.36/240/297.5 | The Middle Urals, mountain-forest zone, temperate continental climate, −14.5/17.8 | Middle Urals, Tagil megazone; harzburgites, serpentinites, basalts, green schists, mica-quartz and graphite-quartz schists, diorites, gabbros, andesites, dacites of the Pre-Jurassic formations; eluvial and diluvial sediments, clays, sandy loams, alluvial sediments of floodplain terraces, pebbles, sands, and loams of the Upper Pliocene and Holocene. | Ferrous and nonferous metallurgy, coking, machinery, chemical industry, production of building materials |
| Magnitogorsk, 0.42/297/392.4 | The South Urals, steppe zone, harsh continental climate, −14.1/19.2 | South Urals, West Magnitogorsk zone; trachibasalts, trachiriolites, basalts, andesites, rhyodacites, lavas, and clastolavas of the Prejurasic formations; alluvial sediments of floodplains, clays, sands, peat, diluvial sediments, eluvial-diluvial sediments, and limes of the Upper Pliocene and Holocene. | Ferrous metallurgy, metal processing, gas-fired power plant |
| Nizhniy Novgorod, 1.3/276/460 | Broad-leaved forests, mixed forests and taiga zone, humid continental climate, −8.9/19.4 | Volga-Ural Anteclise, Pre-Quarternary clays with interbeds of siltstone, sand with gravel of sedimentary rocks, siltstone, loam, marl, gypsum, limestones, and dolomites; alluvial sediments, sands with gravel, loam, clay, eluvial and solifluction formations, sand, eluvial and diluvial Holocene formations. | Machinery, river shipping |
| Rostov-on-Don, 1.1/285/354 | Steppe zone, temperate continental climate, −3/23.4 | East European plate, Rostov ledge; sands, clays, gravel, and pebbles of the Lower Pliocene;limestones, shells, siltstones, and marls of the Upper Miocene; alluvium floodplain terraces, sands, pebbles, loams, sandy loam, eluvial and proluvial sediments of the Upper Pliocene and Holocene. | Machinery, river shipping, food industry |
| Murmansk, 303.8/321/154.4 | Arctic tundra zone, atlantic-arctic temperate climate, −10.1/12.8 | Murmansk megablock represented by Archean granitoids. Pyroxene diorites, tonalites-plagiogranites, magmatite-plagiogranite amphibole, metamorphosed gabbros, diorites, granites, gneisses, biotite amphibolites, magnetite quartzites of the Pre-Jurassic. Declivial marine sediments: sandy silts, mixed-grained sands. | Machinery, shipping, metalworking, food industry, coal-fired power plant |
| Ekaterinburg, 1.387/302/486 | The Middle Urals, forest zone, temperate continental climate, −12.6/19 | Middle Urals, low mountains and hilly plains along the Iset River. Serpentinites, granites, gabbro, diorites, tuffs, tuff sandstones, siliceous and carbonaceous-siliceous shale, quartzite of the Pre-Jurassic; eluvial and diluvial sediments, clays, loams, alluvial sediments of floodplain terraces of the Holocene. | Metal processing, machinery, gas-fired power plant |

### 2.2. Sample Collection

The USDS samples were collected on an irregular grid of at least 40 sampling sites in each city. The sampling site represents the courtyard area of the residential quarter with multi-story buildings. Each sample was taken from the local depressions of the microrelief from 3–5 localizations on the territory of the courtyard space of the quarter. The sample collection procedure was described in detail in previously published papers [22,25,46]. The sample mass was 1–1.5 kg. During the sample collection process, a questionnaire was filled for each sampling site containing information about the conditions of sediment formation, their thickness, the approximate area of the quarter, the proportion of landscaped functional zones, sidewalks, parking lots in the quarter, the quality of cleaning, carrying out construction work, and the approximate time of development of the territory.

### 2.3. Particle Size Analysis

Large roots, stones, debris, and foreign inclusions (glass, plastic, etc.) were removed from the samples. The samples were dried at room temperature. The dried sample was crushed manually using a rubber-tipped pestle, and thoroughly mixed. A representative subsample of about 200 g for particle size analysis was taken from each sample by quartering. To conduct particle size analysis, at least 5 samples were randomly chosen from 40 samples collected in each city.

The special separation procedure was used to determine the granulometric composition and to obtain the solid material of the various particle size fractions of the samples. The technique based on decantation and wet sieving of the material of subsample of 200 g was earlier described in detail by Seleznev and Rudakov [46]. The subsample of 200 g was fractionated into 6 granulometric subsamples with sizes: >1 mm, 0.25–1 mm, 0.1–0.25 mm, 0.05–0.1 mm, 0.01–0.05 mm, and 0.002–0.01 mm. The resulting granulometric subsamples were weighed. The mass fraction of each particle size fraction in the sediment sample was calculated.

### 2.4. Mineral Analysis

The analysis of the heavy mineral concentrate (Schlich analysis) of sediment was used to determine the particles of technogenic origin. Manual analysis was performed for 0.1–0.25 and 0.25–1 mm granulometric subsamples. The fraction of anthropogenic particles was calculated in 0.1–0.25 and 0.25–1 mm fractions. The analytical procedure is described below.

The solid material of the studied granulometric subsample was poured on paper and thoroughly mixed. Then a cone pile was formed from the poured loose material. After that, the material was flattened into a disk 1–2 mm thick. This disk was divided radially into quarters; two opposite quarters were taken for the further analysis of the subsample and the other two were discarded. Such a procedure of quartering and reducing the volume of the material of the granulometric subsample was repeated multiple times until the subsample of the desired weight or volume was obtained. The final volume of the quartered granulometric subsample was approximately 15 mL. Using a blade, the quartered granulometric subsample was distributed on the slide in three parallel lines. To identify and count particles, the lines were formed narrow and sparse. All manipulations with the grain mounts were conducted manually using the binocular microscope. Manipulation with the cone, disk, and the lines of particles, as well as quartering was performed using a wooden stick or copper needle.

The identification of the technogenic particles was carried out by morphology, structure, color, density, optical and physical properties (shape and crystal habitus, splinters, fracture, transparency, luster, elasticity, and hardness). Each particle was photographed using a Carl Zeiss Axioplan 2 optical microscope and binocular microscope equipped with an Olympus C-5060 camera. The size of particles was determined by a calibrated stage/objective micrometer (1 mm divided into 100 units) measurement scale of the optical microscope and its software.

All the particles of the quartered subsample were distributed by type; the fraction of particles of each type was counted.

After quartering and heavy mineral concentrate analysis 2–5 visually typical particles were selected from the part of granulometric subsample attributed to the technogenic phase. These particles were analyzed with a JEOLJSM-6390LV scanning electron microscope equipped with Oxford Instruments INCAEnergy 350 X-Max 50 energy-dispersive spectrometer. At least one image was obtained from the surface of each selected particle. The homogeneity of the chemical composition of the particle surface was identified visually by the color of the image. At least one spectrum of elemental composition was determined for a particle with a flat surface, characterizing its uniform composition. For particles with a concave or convex surface at least two spectra of elemental composition were taken from the surface (in the center of the surface and at its peripheral). For particles with visually different chemical compositions (different shades of gray in the image), at least one spectrum in each light area was taken. For particles with inclusions at least one spectrum was taken on each inclusion, and the linear size of the inclusion was measured. Similarly, at least one spectrum was taken on each area of the external contamination of particles (if it was present). Optical analysis, photography, and scanning electron microscopy (SEM) were carried out in the "Geoanalyst" Center for Collective Use at the Institute of Geology and Geochemistry of the Ural Branch of the Russian Academy of Sciences.

The origin of the particles (technogenic or natural) was finally determined according to the results of their visual analysis (color, luster, morphology, and size) and SEM investigations (surface morphology and chemical composition).

## 3. Results

The number of USDS samples collected in the cities and analyzed fortechnogenic phase is shown in Table 2. The analysis of heavy mineral concentrate was performed in 85 granulometric subsamples of 0.1–0.25 mm and 80 subsamples of 0.25–1 mm in size. For the particle size fraction of 0.1–0.25 mm, 11,985 particles were analyzed with the optical method, and 2306 of them were visually identified as technogenic. For subsamples of 0.25–1 mm in size, 10678 particles were inspected with a binocular microscope, of which 1409 particles were attributed to the technogenic phase.

**Table 2.** The number of urban surface-deposited sediment (USDS) samples collected in the cities and analyzed for technogenic phase.

| City | Number of Samples for Particle Size Analysis | Number of Obtained Particle Size Subsamples, in Which Technogenic Particles Were Selected * | |
|---|---|---|---|
| | | Fraction 0.1–0.25 mm | Fraction 0.25–1 mm |
| Ekaterinburg | 6 | 5 | 6 |
| Magnitogorsk | 10 | 10 | 10 |
| Murmansk | 10 | 10 | 10 |
| Nizhniy Novgorod | 8 | 8 | 7 |
| Nizhniy Tagil | 11 | 11 | 11 |
| Perm | 5 | 5 | 3 |
| Rostov-on-Don | 9 | 7 | 9 |
| Tyumen | 7 | 7 | 5 |
| Ufa | 12 | 12 | 10 |
| Chelyabinsk | 10 | 10 | 9 |

* The subsample was quartered.

The statistical parameters of the fractional distribution of technogenic particles in the surveyed cities in particle size fractions of 0.1–0.25, 0.25–1, and combined fraction of 0.1–1 mm are shown in Figures 2 and 3.

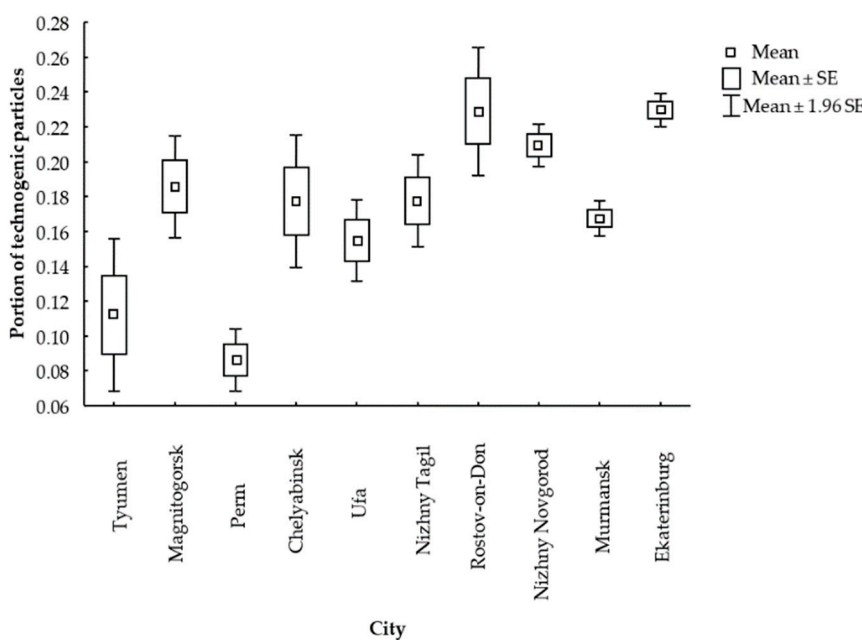

**Figure 2.** The proportion of technogenic particles in the cities in the particle size fraction of 0.1–1 mm (SE—standard error).

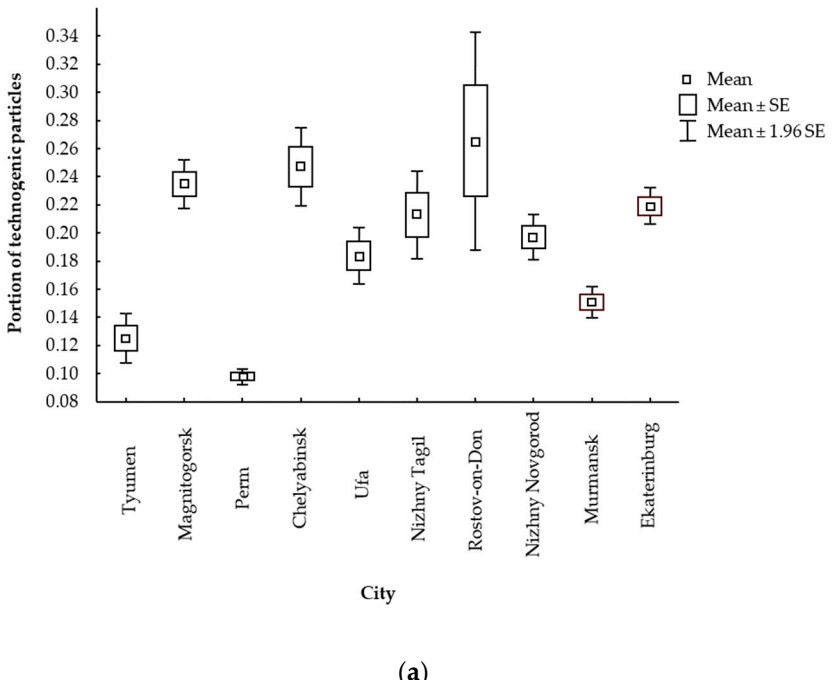

(**a**)

**Figure 3.** *Cont.*

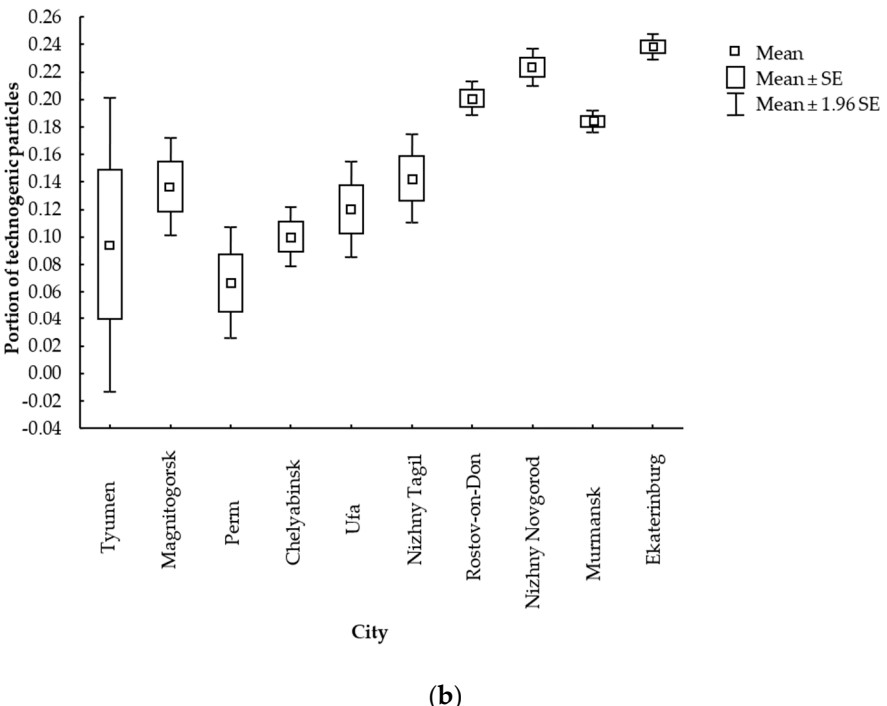

**(b)**

**Figure 3.** The proportion of technogenic particles in the cities: (**a**) in the particle size fraction of 0.1–0.25 mm, (**b**) in the particle size fraction of 0.25–1 mm (SE—standard error).

The additional parameters of distribution (kurtosis, skewness, and the coefficient of variation) of the proportion of technogenic particles in particle size fractions in cities are represented in Tables 3 and 4. The type of distribution of the proportion of technogenic particles in the samples for all cities and fractions is close to unimodal and lognormal. The proportion of technogenic fraction in the samples varies in the range from 0.01 to 0.43, the average value is 0.18. The average coefficient of variation (CV) of the portion of technogenic phase equal to 38% exhibits a wide range of data on the amount of technogenic phases in coarse particle size fraction of the USDS samples. The values of skewness are insignificant, and kurtosis is positive. The values of CV >20% were found in particle size fractions of 0.1–0.25, 0.25–1, and 0.1–1 mm in all cities besides Nizhny Novgorod, Murmansk, and Ekaterinburg.

When analyzing the individual particle size fractions of 0.1–0.25 and 0.25–1 mm (Table 3) and combined fraction of 0.1–1 mm by the cities (Table 4), the proportion of technogenic fraction exceeds 20% for all cities except Nizhny Novgorod, Murmansk and Ekaterinburg. The maximum proportion of the technogenic fraction equal to 0.43 is observed in Rostov-on-Don in the 0.1–0.25 mm fraction. The minimal portion was found in the 0.25–1 mm fraction in Tyumen. The amount of technogenic particles in granulometric fractions of 0.1–0.25 and 0.25–1 as well as in the combined fraction 0.1–1 mm varies significantly between different cities. The largest proportion of technogenic material was found in the USDS samples in Rostov-on-Don, Nizhniy Novgorod, and Ekaterinburg. The parameters of the distribution of portion of technogenic phase in urban areas in the combined particle size fraction of 0.1–1 mm are as follows: arithmetic mean 0.18, geometric mean 0.16, median 0.19, min–max 0.01–0.43, SD 0.07, CI (-/+) 0.06/0.08, CV 38.52 %, skewness 0.21, and kurtosis 1.23.

**Table 3.** The additional parameters of distribution of the proportion of technogenic particles in particle size fractions of 0.1–0.25 and 0.25–1 mm in cities.

| City | Particle Size Fraction, mm | Kurtosis | Skewness | Coefficient of Variation, % | Min/Max |
|---|---|---|---|---|---|
| Tyumen | 0.1–0.25 | −0.79 | −0.87 | 19.01 | 0.09/0.15 |
| | 0.25–1 | 4.51 | 2.09 | 129.91 | 0.01/0.31 |
| Magnitogorsk | 0.1–0.25 | 0.30 | 1.08 | 11.94 | 0.21/0.29 |
| | 0.25–1 | −1.11 | −0.62 | 41.88 | 0.04/0.2 |
| Perm | 0.1–0.25 | 0.90 | 0.97 | 6.61 | 0.09/0.11 |
| | 0.25–1 | 0.00 | 1.24 | 53.83 | 0.04/0.11 |
| Chelyabinsk | 0.1–0.25 | 5.41 | 2.10 | 18.12 | 0.2/0.36 |
| | 0.25–1 | 0.40 | −0.72 | 32.88 | 0.04/0.14 |
| Ufa | 0.1–0.25 | 1.18 | −0.94 | 19.11 | 0.1/0.23 |
| | 0.25–1 | 0.82 | 0.76 | 46.81 | 0.04/0.24 |
| Nizhniy Tagil | 0.1–0.25 | 0.04 | 0.93 | 24.75 | 0.15/0.31 |
| | 0.25–1 | −1.23 | −0.09 | 37.83 | 0.06/0.22 |
| Rostov-on-Don | 0.1–0.25 | −0.80 | 1.22 | 39.36 | 0.19/0.43 |
| | 0.25–1 | −1.19 | 0.29 | 9.33 | 0.17/0.23 |
| Nizhniy Novgorod | 0.1–0.25 | 1.12 | −1.37 | 11.72 | 0.15/0.22 |
| | 0.25–1 | 0.80 | −1.30 | 8.18 | 0.19/0.24 |
| Murmansk | 0.1–0.25 | 0.02 | −0.62 | 11.66 | 0.12/0.18 |
| | 0.25–1 | 3.25 | 1.38 | 6.85 | 0.17/0.21 |
| Ekaterinburg | 0.1–0.25 | −2.01 | −0.40 | 6.75 | 0.2/0.24 |
| | 0.25–1 | −2.52 | 0.15 | 4.82 | 0.23/0.25 |

**Table 4.** The additional parameters of distribution of the proportion of technogenic particles in particle size fraction of 0.1–1 mm in cities.

| City | Curtosis | Skewness | Coefficient of Variation, % | Min/Max |
|---|---|---|---|---|
| Tyumen | 3.40 | 1.40 | 69.04 | 0.01/0.31 |
| Magnitogorsk | 0.15 | −0.78 | 35.91 | 0.04/0.29 |
| Perm | 0.67 | −1.38 | 29.70 | 0.04/0.11 |
| Chelyabinsk | −0.52 | 0.22 | 47.70 | 0.04/0.36 |
| Ufa | −0.73 | −0.41 | 35.72 | 0.04/0.24 |
| Nizhniy Tagil | 0.25 | 0.18 | 35.57 | 0.06/0.31 |
| Rostov-on-Don | 4.38 | 2.33 | 32.83 | 0.17/0.43 |
| Nizhniy Novgorod | 0.93 | −0.97 | 11.65 | 0.15/0.24 |
| Murmansk | 0.37 | −0.36 | 13.52 | 0.12/0.21 |
| Ekaterinburg | −0.04 | −0.43 | 6.94 | 0.2/0.25 |

According to SEM analysis, the studied technogenic particles were divided into types presented in Table 5.

Table 6 shows the morphological features of the various types of particles. Totally 464 particles were analyzed by SEM. The number of particles investigated by cities was: Ekaterinburg 151, Magnitogorsk 22, Murmansk 31, Nizhny Novgorod 127, Nizhny Tagil 30, Rostov-on-Don 71, Tyumen 22, Ufa 9, and Chelyabinsk 1. The chemical composition of the surfaces of various types of particles (without inclusions) is shown in Table 5 as well.

**Table 5.** The chemical composition of the surfaces of various types of particles without inclusions according to scanning electron microscopy (SEM) analysis.

| Type of Particle | Elements | Composition, Mass Portion of Element, % |
|---|---|---|
| Lithoid slag | major | O (31%), Si (21%), C (15%), Fe (10%), Ca (9%), Al (6%), |
| | impurities | Mg (3%), Na (3%), K (2%) |
| Granulated slag | major | O (39%), Si (18%), Fe (15%), Ca (9%), |
| | impurities | Mg (4%), Al (4%), C (2%), Ti (1%), S (1%), K (1%) |
| Iron microsphere (magnetic) | major | Fe (69%), O (24%), |
| | impurities | Si (2%), Ca (1%) |
| Silicate microsphere | major | O (39%), Si (23%), Ca (12%), Fe (8%), Mg (5%), |
| | impurities | Al (4%), Na (2%), Cu (2%) |
| Brick | major | O (35%), Si (22%), Fe (17%), Ca (11%), |
| | impurities | K (3%), Al (3%), Na (2%), Ti (2%), C (2%) |
| Paint | major | O (39%), Ca (15%), Fe (14%), Si (13%), Pb (5%), |
| | impurities | Ti (4%), Mg (3%), Al (3%), K (1%), C (1%), Cr (1%) |
| Glass | major | O (35%), Si (28%), Fe (9%), Ca (8%), |
| | impurities | Al (4%), Cu (3%), Mg (2%), Na (2%), K (1%), Cr (1%) |
| Plaster fragment | major | O (36%), Ca (29%), Si (11%), Fe (6%), C (6%), |
| | impurities | Mg (3%), Al (3%), Na (1%), S (1%), K (1%), Cr (1%) |
| White-coated plaster | major | Ti (46%), O (18%), Ca (15%), Cu (11%), |
| | impurities | Ba (3%), Fe (1%), Al (1%), S (1%) |
| Paint coated plaster | major | Ca (55%), O (30%), |
| | impurities | Si (3%), Ti (3%), C (3%), Fe (2%), Al (1%), Pb (1%) |

**Table 6.** Morphological features of types of technogenic particles in the studied cities.

| Type of Particle | Morphological Features | Size, mm | Possible Origin |
|---|---|---|---|
| Granulated slag | Glassy structure, shell-like breakage, poorly rounded, black, dark brown, dark green, grey, light yellow or colorless, transparent or translucent | 0.3–1 | Metallurgy |
| Lithoid (stone-like) slag | Stone-shaped particles, with a porous structure, crystallized, medium rounded, grey, dark brown, dark green, translucent or opaque | 0.3–1 | Metallurgy |
| Iron microsphere (magnetic) | Spheres, with a smooth or polygonal textured surface, steel-grey, often with thin films of iron oxides, opaque | 0.1–1 | Metallurgy |
| Silicate microsphere | Spheres, sometimes slightly flattened or deformed; the surface is corroded, with cavities and visible cracks; black, dark brown; opaque or colorless translucent with a strong glassy luster | 0.45–1 | Combustion of high ash raw material |
| Brick | Well or completely rounded debris (quartz, clay material, whitewash); red-brown, dark red with inclusions, opaque | 0.5–1 | Construction materials |
| Plaster | Thin, flattened particles, highly fragile; light grey, white, opaque, matt | 0.5–0.8 | Construction materials |
| Glass | Glassy, poorly or perfectly rounded; colorless, yellow, blue, green, transparent | 0.5–1 | Household waste |
| Paint | Thin, flattened, elastic particles; yellow, red, blue, green, with a matt or shiny surface | 0.25–1 | Construction materials |
| Car tires | Smooth particles, high elasticity; black, opaque, matt | 200–1000 | Automobile nonexhaust emissions |

The distribution of different types of technogenic particle in urban areas in the 0.1–1 mm grain size fraction and 0.1–0.25 and 0.25–1 mm fractions are shown in Figures 4–6.

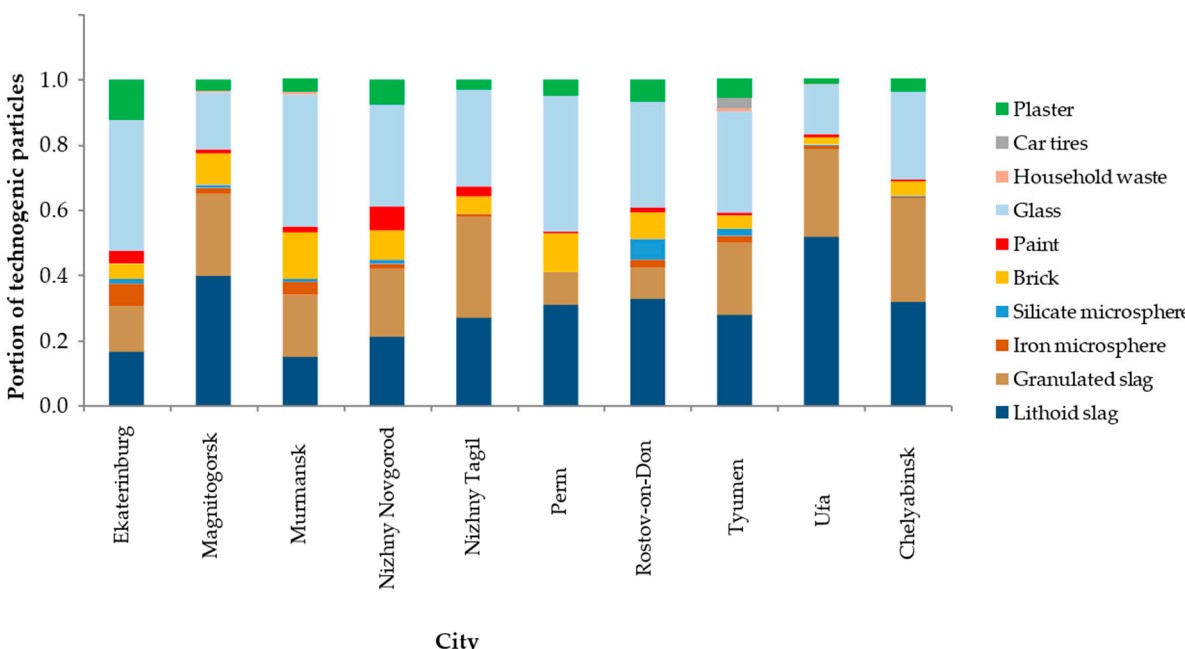

**Figure 4.** Distribution of the amount of different types of technogenic particle in the cities in the granulometric fraction of 0.1–1 mm.

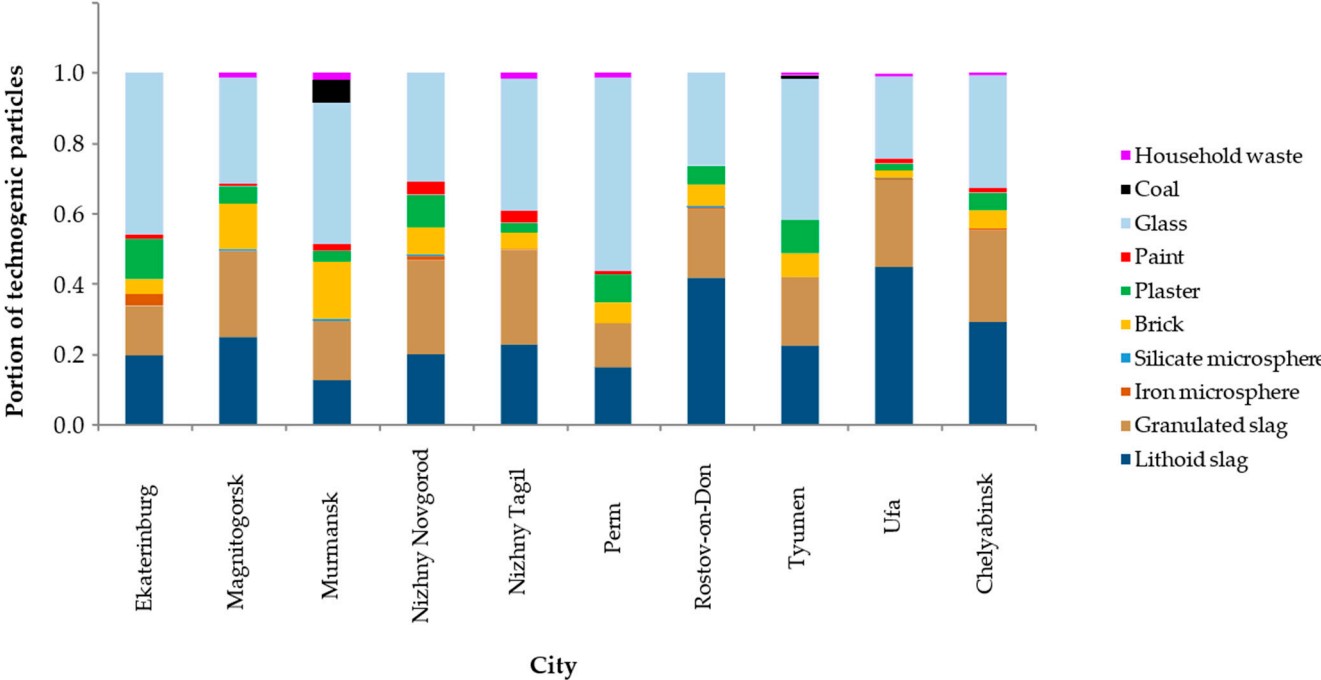

**Figure 5.** Distribution of the amount of different types of technogenic particle in the cities in the granulometric fraction of 0.1–0.25 mm.

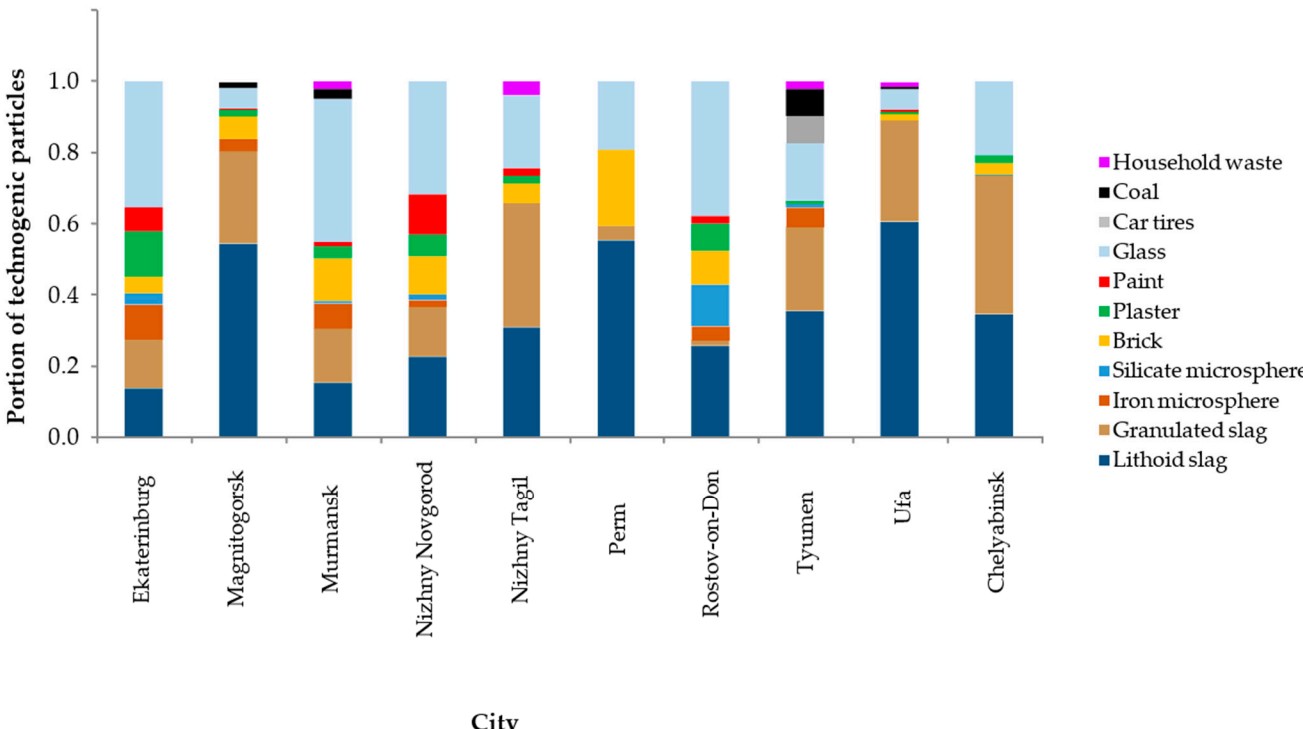

**Figure 6.** Distribution of the amount of different types of technogenic particle in the cities in the granulometric fraction of 0.25–1 mm.

## 4. Discussion

The USDS samples were collected in 10 large cities located in different geographic and climatic zones, and in territories with different geological setting, anthropogenic pressure, and economy. The research was carried out according to the uniform methodology in all the studied cities. A part of the obtained particle size subsamples of 0.1–0.25 and 0.25–1 mm in size did not have enough material to conduct the analysis of heavy mineral concentrate, thus these subsamples were rejected from the technogenic particle investigations. In the cities of Perm, Ekaterinburg, and Tyumen a smaller number of USDS samples were collected, thus a correspondingly smaller number of subsamples for the analysis of heavy mineral concentrate were selected. Such a homogeneous distribution of the USDS sample amount and particle size subsamples did not affect the results of the analysis of heavy mineral concentrate and was suitable for the current study.

The total number of the studied samples is sufficient to assess the contribution of the technogenic component to the USDS solid coarse fractions of 0.1–0.25 and 0.25–1 mm in size. According to the visual mineral analysis, 19% and 13% of particles were characterized as technogenic in particle size fractions of 0.1–0.25 and 0.25–1 mm, respectively. The rest of the particles is represented by the mineral and natural organic fragments.

The proportion of technogenic particles differs from city to city. The largest portion of anthropogenic particles in the USDS coarse fraction was found in Rostov-on-Don, Ekaterinburg, Nizhny Novgorod, Nizhny Tagil, and Magnitogorsk. The high proportion of technogenic particles in these four cities is apparently related to the ferrous metallurgy and mechanical engineering industries. The city of Rostov-on-Don is the most southern of the surveyed cities. According to previous studies, the city has the highest accumulation of dust and USDS due to the arid climate and bad cleaning and management of the urban environment [1,22]. The lower amount of the anthropogenic coarse material was found in Perm and Tyumen. Tyumen is one of the least-polluted cities in Russia, although it has a slightly large number of cars per capita in comparison with other cities [22]. It should be noted that for all cities the proportion of technogenic phase in the combined fraction

of 0.1–1 mm will be consistent with the proportions of the anthropogenic material in the separate fractions of 0.1–0.25 and 0.25–1 mm (Figure 4).

The ratio between the number of technogenic particles visually identified and the total amount of particles in the granulometric subsamples may be used to roughly estimate an error in determining the number of technogenic particles by visual inspection (for subsamples of 0.1–0.25 and 0.25–1 mm, 19% and 13%, respectively).

The SEM-EDS (energy-dispersive spectroscopy) technique allows us to analyze the surface of the particle and determine its chemical composition. Thus this method of analysis is more reliable for the determination of particle type than visual diagnostics. Visual inspection depends on the qualification, physical abilities, and experience of the operator. Therefore, optical methods of research do not fully guarantee the reliability of determination of the particle type. Fully reliable determination of the particle type by its visual features is unattainable and is not required. However, the combination of methods of analysis of heavy mineral concentrate and visual diagnostics is a suitable and easy technically realized procedure to discriminate technogenic particles in comparison with SEM-EDS analysis that requires the investigator to have skills in electron microscopy. At the same time, the analysis of heavy mineral concentrate provides the search of the required particles among the big amount of the similar objects and a rough estimation of the quantity of the objects of interest.

Various types of technogenic particles differ in shape and physical characteristics as well as in chemical composition. The major elements forming the composition of the particle core were O, Si, Fe, Al, Ca, Ti, etc. The minor elements found on the surfaces of the particles and forming the impurities were Mg, K, Cu, Na, etc. In many cases, impurity elements contribute to the environmental pollution, in particular, the composition of various particles of plaster coated with paint and whitewash includes Pb, Cu, and Cr.

The separate group of the cities of the Ural region with a metallurgical industry (Nizhny Tagil, Chelyabinsk, and Magnitogorsk) can be distinguished among the studied cities. Each city in this group has a large metallurgical plant, coking, and coal power plants. The number of technogenic particles does not differ significantly both in fractions of 0.1–0.25 and 0.25–1 mm separately and in the combined particle size fraction of 0.1–1 mm in these cities. According to the results of previous studies [46], the anthropogenic material in the form of slag is used in such cities as a building material, for example, instead of sand and stone in pavement and road construction in residential areas. There is also a coal power plant in Murmansk. It can be assumed that in the group of four cities, technogenic particles, in particular slag, can enter the USDS material with emissions from power plants and smelters.

All the studied cities have a high automobile traffic network, as well as road construction works being underway. The technogenic components (especially fly ash) are often used as construction materials or backfill materials on pavements. Such material can be transferred into the USDS by the wheels of vehicles in the residential area. In general, the amount of technogenic material is comparable to the data obtained for other cities [15].

The distribution of the proportion of technogenic particles in the samples deviates from the normal and is close to lognormal and asymmetric. Several studies conclude that the lognormal distribution of elemental concentrations in environmental compartments or close to it relates to additional anthropogenic input of the elements [48–50]. In our study, the conclusion about the distribution of the proportion of anthropogenic particles in the studied samples close to lognormal was expected; however, it is important to take into account the uncertainty of information about the source of technogenic particles in the urban environment. The coefficient of variation of the portion of anthropogenic particles also confirms the fact of the heterogeneity of the sample populations in the studied cities.

The analysis of the technogenic phase composition of USDS samples in the combined fraction of 0.1–1 mm shows that slag particles predominate in all cities and, besides, a large amount of domestic wastes (glass), the particles of construction materials (plaster and brick), and to a lesser extent paint particles, are observed. The analysis of the distribution of

the number of technogenic particles in fractions of 0.25–1 and 0.1–0.25 in combination with the results of analysis of heavy mineral concentrate allows concluding that technogenic particles in finer fraction of 0.1–0.25 mm may be the result of the destruction of larger particles from particle size fraction of 0.25–1 mm.

The individual particle size subsamples reveal the features of the cities that may be related to the contribution of the studied types of technogenic particle to the city pollution. For example, the granulometric fraction of 0.25–1 mm in Tyumen contains about 10% of coal, which indicates the presence of local coal-fired boilers in addition to the main stationary gas-fired power plants in the city. Moreover, the residential neighborhoods with multi-story buildings in Tyumen are adjacent to low-rise wooden buildings, where heating is provided from coal combustion [51]. Tyumen also has approximately 8% of tire material in fraction of 0.25–1 mm, indicating a high number of cars per capita (higher than in other cities). In Murmansk, with a coal cargo port located within the city center, about 7% of coal is found in particle size fraction of 0.1–0.25 mm.

The elemental composition of technogenic particles is formed by different elements depending on the particle origin. Major elements may include the same elements that form the mineral component of the urban sediment: Si, Al, Ca, Fe, Mg, etc. [23]. However, each type of anthropogenic particle relates to some source of environmental pollution and to a related potentially harmful elements. In the current study, the granulometric subsamples were obtained after washing the samples with distilled water and, therefore, minor element content in the studied technogenic particles refers to trace elements rather than to material adsorbed on the particle surfaces. The accumulation of paint particles and colored plaster debris in the USDS contributes to the pollution of the urban environment with potentially toxic elements. The technogenic particles in the USDS samples tend to the formation of the geochemical anomalies in the urban area and increased concentrations of heavy metals in contemporary surface sediments.

The uncertainties in this study are related to the following factors:

- the errors of the operator in identifying the particle type;
- particle loss in particle size analysis under water washing and decantation;
- counting errors in the analysis of heavy mineral concentrate;
- the location of sampling sites in residential blocks far from roads, etc.

Taking into account the sources of uncertainty, the obtained results satisfactory characterize the anthropogenic component of the surface sediments in residential areas in large Russian cities.

The total amount of the USDS estimated for several Russian cities varies in the range from 1.8 to 3.2 kg/m$^2$ including approx. 65% of fraction >100 μm [23,24]. Thus, the amount of anthropogenic material in Russian cities varies from 0.21 to 0.37 kg/m$^2$. This result shows a quite large accumulation of technogenic material in the urban environment.

The preliminary analysis of microplastic particles in the USDS samples in Russian cities allowed the amount of microplastic particles <1 mm to be considered insignificant in this environmental compartment [28]. The results of the assessment of the number of microplastics are not presented in the current paper; however, further studies may use the methodological approaches represented in the paper to search for plastic microparticles and estimate their amount.

## 5. Conclusions

The combined approach was applied to assess the number of technogenic components in loose coarse sedimentary material in an urban environment. When determining the types of technogenic particle, the shape of the particles as well as their color and surface morphology are of great importance. The approach was based on the methods of quantitative and quantitative mineral, SEM-EDS, and environmental analysis. This approach can be implemented in other environmental studies for similar purposes.

The study of technogenic particles in the contemporary anthropogenic sediments allows important information about the sources of pollution to be obtained, especially about

local non-point sources of pollution and their characteristics in an urban area. According to revealed quantitative indicators, it has been shown that the USDS in Russian cities contain a significant part of technogenic particles. Surveyed cities are differentiated by the amount and types of the technogenic particles preferably presented in the local USDS in residential area. Techogenic material may impact the transformation of typomorphic element associations in the urban environmental compartments. The trace elements found among the technogenic particles as impurities may change the microelement composition within the components of the urban sediment cascade.

**Author Contributions:** Conceptualization, methodology, formal analysis, data curation, writing—Original draft preparation, supervision, review and editing, visualization, project administration, funding acquisition, planning of laboratory analysis, A.S.; laboratory analysis, E.I.; field study, writing—Original draft preparation, review and editing, I.Y.; field study, review and editing, G.M. All authors have read and agreed to the published version of the manuscript.

**Funding:** The reported study was funded by Russian Science Foundation, project number № 18-77-10024.

**Institutional Review Board Statement:** Not applicable.

**Informed Consent Statement:** Not applicable.

**Data Availability Statement:** Not applicable.

**Acknowledgments:** The optical, mineral, and SEM-EDS analyses were conducted in the Common Use Center "Geoanalyst" in IGG UB RAS.

**Conflicts of Interest:** The authors declare no conflict of interest.

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
