# Peer review of "Coarse Technogenic Material in Urban Surface Deposited Sediments (USDS)"

_atmosphere, doi:10.3390/atmos12060754_

Round 1

Reviewer 1 Report

Here I would like to indicate issues wich require correction or clarification.

  1. Abbreviation of SEM needs clarification. I can guess that SEM stands for scanning electrom microscopy, but some readers  may not know it.
  2. Line 333: "It can be assumed that the sources of anthropogenic particles are also anthropogenic". It is not clear for me what this sentence means. It seems that it is obvious that anthropogenic sources produce anthropogenic particles.  
  3. RDS - road deposited sediments? 
  4. Line 39: "The large size fractions material of road deposited sediments (> 100 μm) contains heavy metals in the amount similar to the fine fractions". This phrase could be understood incorrectly. If "amount" is concentration, the phrase contradicts the paper [16] as well as other numeros papers focusing on particle size distribution. Finer particles are enriched with metals much stronger than coarse particles. If "amount" is mass of heavy metals within/on particulate matter, then I agree. Although concentration of heavy metals is low on coarse partciles, mass is high due to relatively high mass of coarse particles. 
  5. Line 70: "(references)" - I assume some references should be placed here?
  6.  Line 77: "PM2-PM10" - I guess PM2.5, not PM2. 
  7. Table 1, column "Geological features": There are too many details, but I have not found how they were used in the further text. I would like to recommend to reduce this detalization and keep only most imortant features. 
  8. Line 124:  "Microrayons" - only Russian or post-Soviet readers can understand what it means. Probably it would be good to use more common term, something like: "Sampling was carried out in residential area represented by multi-story buildings."
  9. General commnet to section 2.4. The authors explain what features or parameters of the sampled particles and what methods were used to distinguish technogenic particles from other ones. However, it remained unclear for me, how the authors finally decide what partcile is technogenic and what is not. Was it shape, size, some features of spectrum or somethin else? 
  10. Fig. 2.3 - "SE" - should be clarified. 
  11. Table 5. Does "minor element" mean "trace element"? If so, I recommend to use more common term "trace elemet". If not, please clarify what is "minor".  
  12. Line 375: "XX to XX kg/m2". I assume there must be some values?

Author Response

Dear Editor,

Please find attached the carefully revised version of the manuscript.

The manuscript was reworked according to the reviewers’ comments. Language editing was made. Some additions and explanations in the manuscript were made. The Reviewers’ comments were taken into account.

Best wishes,

Andrian Seleznev

Reviewer 2 Report

This study analysed the composition of coarse technogenic material in different cities and it contains enough results. The methods are clear and nicely presented. However, some major concerns are listed below;

  1. The significance and novelty of the study are not clear from the abstract. The authors should revise the abstract with a background sentence and the significance of the study.
  2. It’s not clear from the abstract how this study will improve the knowledge of the field?
  3. The introduction contains enough informations. However, authors should include more recent papers.
  4. The authors need to discuss how this present study is different from ‘some geochemical characteristics of puddle sediments from cities 524 located in various geological, geographic, climatic and industrial zones’ study?
  5. In Table 2, the authors collected 25 samples for a city and analysed 6 samples? What is the reason behind it? Did the authors try more samples? If the authors analyse 25 samples, is it will provide different results? This reviewer believes all samples are not the same and there should be some error if the authors analyse different samples.
  6. In table 5, the authors presented the composition of the different particle. For example, lithoid slag O, Si, C, Fe etc are found major. Isn’t it expected? If yes, what’re the new findings of this study and how it differs from other studies?
  7. From the conclusion, the major findings are not clear and how this study will improve the knowledge of the field.

Author Response

Dear Reviewer,

Please find attached the carefully revised version of the manuscript.

The manuscript was reworked according to the reviewers’ comments. Language editing was made. Some additions and explanations in the manuscript were made. The Reviewers’ comments were taken into account.

Best wishes,

Andrian Seleznev

Round 2

Reviewer 2 Report

The authors tried to improve the manuscript based on the reviewer suggestions. However, some point still needs improvement. Authors replied regarding the recent paper that they cited papers from 2010-2020. 2010 papers are not recent paper and authors should understand this. They should cite all relevant papers from 2016-2021.

Authors need to write a sentence in the introduction section about how the previous study is different from the present study. It will help the readers to understand the significance of this work.

The major findings still not clear from the conclusion. The authors replied that 'see the abstract' and that's not the proper method. The reader will see the conclusions for the final findings. Please make some bullet points for the conclusion section, and it will help the readers to understand the work.

Author Response

Dear Editor and reviewer,

Please find attached the carefully revised version of the manuscript. The manuscript was reworked according to the reviewers’ comments. The following comments were taken into account.

Best wishes,

Andrian Seleznev
